# Diabetes Mellitus and Obesity as Prognostic Factors in Arthroscopic Repair of Chronic Rotator Cuff Tears

**DOI:** 10.3390/jcm12020627

**Published:** 2023-01-12

**Authors:** Javier Álvarez de la Cruz, Marye Mercé Méndez Ojeda, Nuria Álvarez Benito, Alejandro Herrera Rodríguez, Jose Luis Pais Brito, Francisco Jesús Márquez Marfil

**Affiliations:** 1Orthopedic Surgery and Traumatology Service, University Hospital of the Canary Islands, 38320 La Laguna, Spain; 2Orthopedic Surgery Service, Surgery Department, La Laguna University, 38320 La Laguna, Spain

**Keywords:** rotator cuff tear, diabetes, obesity, shoulder arthroscopy surgery

## Abstract

Metabolic diseases such as obesity and diabetes mellitus seem to have an influence on reoperation and long-term functional outcomes after arthroscopic repair of chronic rotator cuff tears. High prevalence of these pathologies can be found in the Canary Islands. A retrospective cohort study was carried out, in which 80 patients undergoing shoulder arthroscopic surgery for the repair of chronic rotator cuff tears were included, with a minimum follow up of 5 years, to study the occurrence of complications, reoperation, and functional outcomes. Functionality after surgery improved in 75% of patients with diabetes and remained the same or worsened in 25% (OR = 1.444). In the group of non-diabetic patients, 83.9% had improved function after surgery while it remained the same or worsened in 16.1% (OR = 0.830). Functionality after surgery improved in 76.6% of obese patients and remained the same or worsened in 23.4% (OR = 1.324). In the non-obese group, 87.9% had improved function after surgery, while it remained the same or worsened in 12.1% (OR = 0.598). Despite not obtaining statistically significant differences, the analysis of the results obtained suggests that obesity and diabetes could act by decreasing the subjective improvement in functionality after surgery, and, in the case of obesity, also increase the risk of reoperation.

## 1. Introduction

Chronic rotator cuff (RC) injury is a common pathology, with a prevalence that ranges from approximately 13% of people over 50 years of age, rising to 25% when reaching 60 years of age, and 50% when over 80 years of age [1,2]. However, it is difficult to determine the real incidence of this pathology, since ruptures are not always symptomatic, and it is possible to estimate a much higher number of cases than those recorded [3].

Although most of these lesions are asymptomatic when the size of the lesion is small, it has not been demonstrated that the size of the lesion is directly related to the symptomatology of the patients [1,4]. If symptomatology does occur, the most common signs are pain, muscle weakness, and decreased function and range of motion of the shoulder joint [4,5].

The appearance of arthroscopic shoulder surgery (ASS) in the 1980s was a revolution in the management of pathologies related to CR tendinopathy. This technique is widely validated by the literature for the repair of ruptures secondary to chronic tendinopathy [6,7,8], with functional results similar to open techniques while generating less morbidity and accelerating hospital discharge and subsequent recovery [9,10].

Several factors have been described that may be determinant in establishing a favorable prognosis in the patient’s recovery after this surgery, including: demographic factors (minority age and male gender), clinical factors (absence of osteoporosis, absence of diabetes, high-performance athlete patient, and complete passive preoperative range of motion), factors related to the integrity of the rotator cuff or to the intervention (repair in the same procedure of the acromioclavicular or biceps joint) [1,3,11] and socioeconomic factors [12]. However, the relationship of diabetes with this pathology and its therapeutic management is not entirely clarified. Diabetes seems to be associated with an increased risk of rupture in chronic RC tendinopathies, as well as with a possible deficit in functional outcomes after arthroscopic tendon repair [13], but the results are inconclusive. High glucose environments seem to compromise the biomechanical properties of the diabetic tendons, as well as to produce relevant histopathological alterations, which eventually result in deficient tendon repair, maintenance, and remodeling [14].

Due to the high rate of obesity and diabetes in the Canary Islands [15], Spain, a retrospective study was conducted to investigate how obesity and diabetes influence the outcomes of tendon repair of the RC by ASS. 

The hypothesis of this study is that both obesity (body mass index (BMI) > 30) and diabetes are a risk factor for both reoperation of the RC after ASS and for worse long-term functional outcomes.

## 2. Materials and Methods

### 2.1. Study Design

This study was a retrospective cohort study.

### 2.2. Study Subjects

The sample consisted of patients who underwent ASS for repair of the RC by expert surgeons of the orthopedic surgery department of a tertiary hospital between January 2010 and December 2015. Patients who underwent surgery during this period were selected in order to be able to assess the occurrence of complications and functional outcomes at least 5 years after the operation up to the time of this study.

### 2.3. Inclusion Criteria

-Patients with chronic rotator cuff tear, diagnosed by ultrasound or MRI.-Follow-up time of at least 5 years.-Intervention performed by expert surgeons in this technique (all of them with similar learning curves: all patients included in the study have been treated by surgeons with more than 10 years of experience in arthroscopic shoulder surgery techniques, all of them having completed more than 200 cases of this type).

### 2.4. Exclusion Criteria

-Absence of follow-up. -Non-acceptance of verbal consent via telephone call to take part in the study.-Subacromial impingement diagnosed by ultrasound or MRI (patients with osseous or ligamentous radiologic factors that could be responsible for the rotator cuff tear, such as: Bigliani Type III acromia, persistent os acromiale, acromial enthesophytes secondary to acromioclavicular osteoarthritis, coracoacromial ligaments with >1 bundle).-Diagnosis of depression or other psychiatric disorders.

### 2.5. Research Variables

-Date of birth.-Age (years).-Sex (male or female).-Obesity (yes/no). Patients with a body mass index greater than 30 were included in the obese group.-Diabetes (yes/no). Patients with type II diabetes mellitus at the time of the operation were included in the group of diabetic patients (two separated test of HbA1c > 6.5%).-Smoking (yes/no).-Dominance (right/left).-Side operated on (right/left).-Date of surgery.-Age at the time of surgery (years).-Complications (yes/no): specifying which (postoperative organ pathology, infection, wound dehiscence, etc.)-Reintervention (yes/no): specifying cause and date of reintervention.-Residual pain (yes/no).-Pain improvement or worsening after surgery.-Subjective functionality after surgery (improvement, same, or worsening).-Subjective comparison of functionality with contralateral shoulder expressed as a percentage. A total of 100% corresponded to complete functionality, without restrictions and without differences with respect to the healthy shoulder; and 0% corresponded to an extremely limited functionality with respect to the healthy shoulder, not being able to incorporate it into the basic activities of daily living (BADL).-Satisfaction level expressed by the answer to the following question: Would you have the surgery again? (yes/no) A situation was considered in which it would be necessary to operate again, and it was asked if you would be willing, based on your previous experience, to be operated on again or not.

### 2.6. Data Collection

The study was approved by the research ethics committee of the third-level hospital with protocol code: CHUC_2021_73. The data collection was carried out between September 2020 and June 2021. All patients met the inclusion criteria and no exclusion criteria were identified. The data collection notebook was completed, in which no personal data appeared, only the anonymized code assigned for the study (PAC-001, PAC-002, etc.). The basic information on the patients (demographic data, comorbidities, and type of intervention) was obtained from the hospital’s computerized clinical records. The data presented in this study are openly available in Figshare at https://doi.org/10.6084/m9.figshare.21405144.v1 (Dataset posted on 26 October 2022).

### 2.7. Statistical Analysis of Data

The information obtained was stored in the logbook in Microsoft Excel and SPSS platforms for further statistical analysis. Comparisons of proportions were performed using the chi-square or Fisher’s exact test, as appropriate. Group comparisons of quantitative and ordinal variables were performed with Student’s *t*-tests, Mann–Whitney, or ANOVA, as appropriate. Logistic regression analysis was used for categorical dependent variables. The difference in functionality with the contralateral shoulder expressed as a percentage, depending on the presence of diabetes or obesity, was tested using the Mann–Whitney U test for two independent samples, after assuming that the functionality variable does not present normality but homogeneity between the variances in the diabetes categories. Probability values of less than 0.05 were considered significant. Data analysis was performed with the SPSS statistical package (IBM Corp., released 2017; IBM SPSS Statistics for Windows, Version 25.0. Armonk, NY, USA).

## 3. Results

The study included a total of 80 patients, 51 women (63.75%) and 29 men (36.25%), who underwent arthroscopic rotator cuff repair between the ages of 24 and 80. The mean age at the time of surgery was 57.92 ± 10.52 years, with the largest group being between 60 and 68 years of age with 33.75% of the patients studied (n = 32) (Table 1). 

The dominant hand of the patients was the right hand in 97.5% (n = 78), and the side most frequently operated on was the right hand with 71.3% (n = 57) (Table 2).

The occurrence of complications was 10% (n = 8), of which 87.5% (n = 7) had to be reoperated on. The presence of residual pain after surgery was observed in 50% (n = 40) of the patients and pain after surgery compared to previous pain improved in 88.8% (n = 71) (Table 3).

Shoulder function after surgery improved in 81.3% (n = 65) and remained the same or worsened in 18.8% (n = 15). A total of 81.3% (n = 65) reported being satisfied with the results of the operation, the management of the operation by the hospital service, and would be willing to undergo surgery again.

### 3.1. Impact of Diabetes

The group of diabetic patients represented 30% (n = 24) of the sample, of whom 91.7% (n = 22) had no complications, while 8.3% (n = 2) did (OR = 0.818, 95% CI = 0.234–2.855). In the non-diabetic group of patients, representing 70% (n = 56), 87.5% (n = 49) had no complications while 10.7% (n = 6) did (OR = 1.080, 95% CI = 0.704–1.658).

Of the total number of re-operated patients [8.8% (n = 7)], 100% of patients did not have diabetes (OR = 1.490, 95% CI = 1.269–1.749).

Assessing the presence of residual pain, 54.16% (n = 13) of diabetic patients had pain (OR = 1.182; 95% CI = 0.603–2.316) while 48.21% (n = 27) of non-diabetic patients had residual pain (OR = 0.931; 95% CI = 0.698–1.241).

Functionality after surgery improved in 75% (n = 18) of patients with diabetes and remained the same or worsened in 25% (n = 6) (OR = 1.444, 95% CI = 0.693–3.009). In the group of non-diabetic patients, 83.9% (n = 47) had improved function after surgery while it remained the same or worsened in 16.1% (n = 9) (OR = 0.830, 95% CI = 0.535–1.288). 

To assess the difference in function compared with the contralateral shoulder as a dependence of the presence of diabetes, the Mann–Whitney U test was used (Mann-Whitney U = 518.000; Z = −1.642; *p*-value = 0.101) (Figure 1).

### 3.2. Impact of Obesity

The obese group of patients undergoing surgery accounted for 58.8% (n = 47) of patients, of whom 89.4% (n = 42) had no complications (OR = 1.071, 95% CI = 0.605–1.897). In the group of non-obese patients [41.2% (n = 33)], 90.9% (n = 30) had no complications (OR = 0.900; 95% CI = 0.353–2.293).

Of the total number of reoperated patients [8.8% (n = 7)], 57.1% (n = 4) were obese (OR = 0.970, 95% CI = 0.497–1.895) (8.51% of patients with obesity), compared with 42.8% (n = 3) without obesity (OR = 1.043, 95% CI = 0.425–2.561) (9.09% of patients without obesity).

Assessing the presence of residual pain, 59.57% (n = 28) of obese patients had pain (OR = 1.238; 95%CI = 0.853–1.797) while 36.36% (n = 12) of non-obese patients had residual pain (OR = 0.737; 95%CI = 0.432–1.256).

Functionality after surgery improved in 76.6% (n = 36) of obese patients and remained the same or worsened in 23.4% (n = 11) (OR = 1.324, 95% CI = 0.910–1.927). In the non-obese group, 87.9% (n = 29) had improved function after surgery, while it remained the same or worsened in 12.1% (n = 4) (OR = 0.598, 95% CI = 0.247–1.444). The percentage of function relative to the non-operated shoulder in the presence of obesity was tested using the Mann –Whitney U test for two independent samples (Mann–Whitney U = 612.500; Z = −1.618; *p*-value = 0.106) (Figure 2).

## 4. Discussion

This study, like previous before [13,14,16], supports the repair of chronic tendinopathy of the RC by arthroscopic techniques, with good long-term results for the most disabling symptoms of this injury: pain and deficit of shoulder function. The high level of satisfaction with the results obtained after the intervention is also noteworthy.

Despite not obtaining statistically significant results, the analysis of the data suggests that both obesity and diabetes could worsen the long-term functional prognosis of those patients who undergo this intervention, with higher rates of residual pain and less subjective improvement in shoulder function after the intervention. However, even taking into account this detriment in the obesity and diabetes groups, we can affirm that the intervention generally achieves few complications and good functional results in both groups.

In addition to anatomical or lifestyle factors, several authors have proposed that vascular impairment of supraspinatus tendon is key for its attrition and degeneration, which eventually may lead to the RC rupture [17,18,19]. For that reason, vascular disease risk factors such as obesity, diabetes, hypertension, or decreased physical activity may contribute to decreased vasculature in these patients [19,20,21].

The role of obesity in arthroscopic shoulder surgery goes beyond a vascular problem, causing an increased complexity in anesthesia, positioning, and portal placement, as well as significantly longer operative times and length of hospitalization [17]. Although this role is yet to be fully clarified, some studies have addressed a negative impact on functional outcomes and increased risk of rupture after repair compared with the control group [17,22,23]. In addition, obese patients have higher mechanical strain on the RC tendons due to arm weight, which may have negative effects as well [23].

Regarding diabetes, people who suffer from it had more than three times the odds of tendinopathy compared with the controls [16]. The chronic hyperglycemic state seemed to produce excessive accumulation of advanced glycation end products (AGEs), altered inflammatory responses, insensitive neuropathy, and neovascularization, which may account for the altered tendon biomechanical, cytological, and molecular properties [24].

Lu P. et al. use the term “diabetic tendinopathy” to refer to these alterations in tendon homeostasis in an attempt to understand the pathogenesis of diabetes in relation to the increased risk of rotator cuff tendon rupture after its repair [14]. Reinforcing this idea, Egemen O. et al., studied in animals the effects of diabetes on tissue regeneration, functionality, and histopathology in tendon injuries, observing a delay in regenerative capacity and recovery of functionality in the presence of diabetes compared with the control group [25]. Since one of the causal agents of the primary rupture does not cease, it is logical to think that its correct healing after surgery will be altered, affecting the functional results and long-term reoperation rate.

Borton Z. et al., in a study with similar characteristics to the one presented, observed up to twice as many complications and re-rupture of the structure after injury repair, and up to four times worse functionality in the group of diabetic patients compared with the control group [26]. Although this study follows the same line as the one presented here, affirming that diabetes worsens the functional prognosis of the surgical management of this pathology, we consider that the magnitude of these effects has yet to be clarified. While, with the results of this study, we could not advise against surgery in diabetic and/or obese patients, studies such as the one by Borton Z. could lead to considering diabetes as a contraindication to this intervention.

This study, like others, is not free of the limitations inherent to retrospective studies, such as the small sample size and the method of data collection, mainly in relation to the subjectivity of pain and functionality. The telephone clinical interview provides convenience and speed in obtaining data. In contrast, physical examination in person by a specialized surgeon could provide more objective results with respect to shoulder functionality, being able to evaluate the degrees of active or passive joint balance, more objective functionality scales such as the DASH score or the Constant score, which would provide results closer to real functionality, although it would also imply greater expenditure of resources and time. Relevant conclusions could be obtained as well by taking into account the presence of diabetes and obesity in the same patient, the size of the tear, the presence of associated shoulder injuries prior to the intervention, or the degree of compliance with the rehabilitation protocol after the intervention.

Finally, it might be interesting, within the group of diabetic patients, to also pay attention to the possible differences between patients with adequate glycemic control after surgery versus those with inadequate control. For this reason, we consider it essential to carry out further studies, preferably prospective and with a larger sample size, to clarify the exact role of both obesity and diabetes in this pathology.

## 5. Conclusions

Arthroscopic repair of chronic rotator cuff injury offers generally good results regardless of the presence of obesity or diabetes. This study cannot affirm that diabetes or obesity, in cases of chronic tendinopathy of the RC, behave as independent risk factors for not presenting subjective improvement in functionality in the long term after arthroscopic repair, or for increasing the risk of reoperation. Despite not obtaining statistically significant differences, the analysis of the results obtained suggests that obesity and diabetes could act by decreasing the subjective improvement in long-term functionality after surgery, and, in the case of obesity, also increase the risk of reoperation.

## Figures and Tables

**Figure 1 jcm-12-00627-f001:**
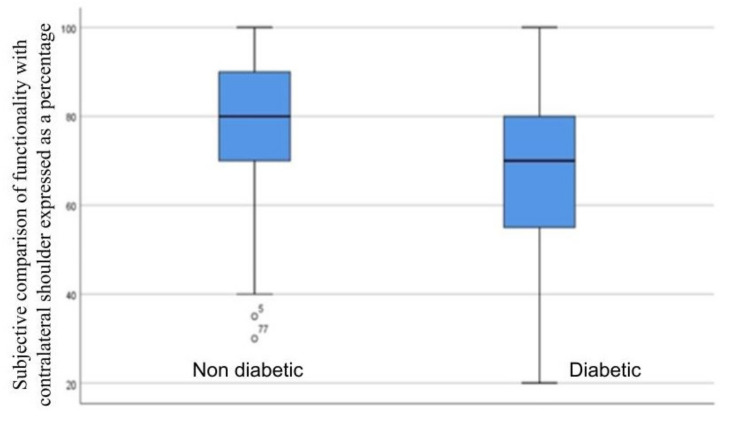
Mann–Whitney U test to compare shoulder function after surgery with the presence of diabetes.

**Figure 2 jcm-12-00627-f002:**
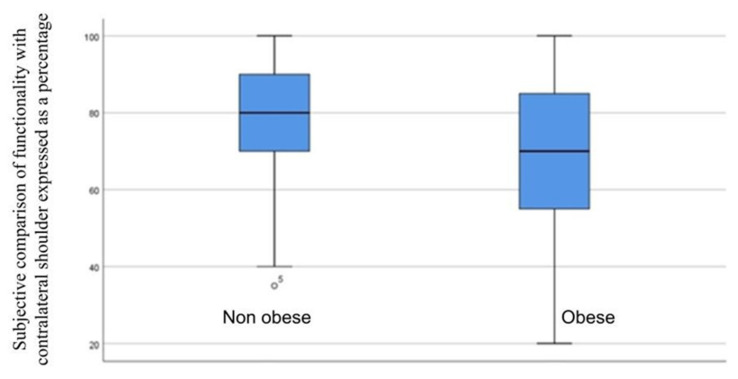
Mann–Whitney U test to compare shoulder function after surgery with the presence of obesity.

**Table 1 jcm-12-00627-t001:** Age expressed in years on the day of the intervention.

Age Range	N	%
20–28	1	1.25
28–36	1	1.25
36–44	5	6.25
44–52	12	15
52–60	22	27.5
60–68	27	33.75
68–76	9	11.25
76–84	3	3.75
Total	80	

Mean age = 57.92 years. Standard deviation = 10.52 years.

**Table 2 jcm-12-00627-t002:** Sex, dominant hand, and side operated.

Sex	Dominant Hand	Side Operated
Male: N = 29; 36.25%	Right: N = 78; 97.50%	Right: N = 57; 71.25%
Female: N = 51; 36.25%	Left: N = 2; 2.50%	Left: N = 23; 28.75%

**Table 3 jcm-12-00627-t003:** Presence of residual pain and need for reoperation in relation to the presence of obesity or diabetes.

		Patients		Residual Pain		Reoperation	
		N	% ^1^	N	% ^2^	N	% ^2^
Diabetes Mellitus	Yes	24	30	13	54.16	0	0
	No	56	70	27	48.21	7	12.5
	Total	80		40		7	
Obesity	Yes	47	58.8	28	59.57	4	8.51
	No	33	41.2	12	26.26	3	9.09
	Total	80		40		7	

^1^ Percentage of total study sample. ^2^ Percentage within the group of the factor under study.

## Data Availability

The data presented in this study are openly available in Figshare at https://doi.org/10.6084/m9.figshare.21405144.v1 (Dataset posted on 26 October 2022).

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
