# Peer review of "Diabetes Mellitus and Obesity as Prognostic Factors in Arthroscopic Repair of Chronic Rotator Cuff Tears"

_jcm, 2023, doi:10.3390/jcm12020627_

Round 1
Reviewer 1 Report
Thank you very much for your study on the influence of diabetes on the outcomes of chronic rotator cuff repair, which I read with great interest.
Prior to publication I would suggest you to consider the following changes:
1. Inclusion criteria: What do you mean by similar learning curves of the surgeons? (e.g. number of operations performed, years of duty in surgery, etc)
2. Exclusion criteria: Subacromial impingement: This describes a complex of different pathophysiological entities. Please refine this definition
3. Research variable: Did you differentiate between different degrees of obesity (e.g. BMI >30, >35, >40?)
4. There are established scores for describing the functionality of shoulder (DASH, ASES, UCLA, Constant, etc). Why did you not use one of these scores?
5. Table 3: Please change Si to Yes
6. Table 2: What was the reason to group patients by age with intervals of 8 years?
7. Did you perform a power analysis? If yes, please present, if not please describe why
8. What was the definition for diabetes at your clinic? HbA1c value?
9. Did you differentiate between diabetes patients that needed insulin and such that did not yet?
10. In your discussion you describe it would be interesting to investigate differences between patients with adequate glycemic control and those without. Is this not possible retrospectively as the certainly went to their general medical doctor for control of Hb1Ac?
11. It is known that degenerative and fatty tendons are difficult to suture. There is even a classification for rotator cuff tears considering this factor: Walter SG, Stadler T, Thomas TS, Thomas W. Advanced Rotator Cuff Tear Score (ARoCuS): a multi-scaled tool for the classification and description of rotator cuff tears. Musculoskelet Surg. 2019 Apr;103(1):37-45. doi: 10.1007/s12306-018-0535-y. Epub 2018 Mar 2. PMID: 29500730.
Reviewer 2 Report
The paper presents a retrospective cohort study, in which 80 patients who underwent arthroscopic shoulder surgery for the repair of chronic rotator cuff tears were included, to study the occurrence of complications, reintervention and functional results.
In principle, the work is properly structured, in the introduction the problem addressed by the study is presented and the research hypothesis is formulated, which could be better highlighted by editing in a separate line.
The materials and methods section is written synthetically, but the data collection section should still indicate the period in which the data collection was carried out.
In the results section, it should be shown if the number of patients included in the study is statistically significant.
In Table 3 - Presence of residual pain, column 2 – “Si” should be clarified/translated.
The discussion section interprets the results of the study in relation to the results of other studies in the field, but in the first paragraph it is not clear which is the previous study referred to "This study, like previous before,..."
I appreciate that it would be useful to introduce a separate paragraph with conclusions of the study in which the readers can have at their disposal the main findings in a synthetic form.
The list of bibliographic references duplicates the serial numbers of the works and must be completely corrected.
Round 2
Reviewer 1 Report
Thanks for revising the manuscript